# CONNECTION STRENGTH-BASED OPTIMIZATION FOR MULTI-TASK LEARNING

## ABSTRACT

The goal of multi-task learning is to learn diverse tasks within a single unified network. As each task has its own unique objective function, conflicts emerge during training, resulting in negative transfer among them. Earlier research identified these conflicting gradients in shared parameters between tasks and attempted to re-align them in the same direction. However, we prove that such optimization strategies lead to sub-optimal Pareto solutions due to their inability to accurately determine the individual contributions of each parameter across various tasks. In this paper, we propose the concept of task priority to evaluate parameter contributions across different tasks. We identify two types of connections to learn and maintain task priority: implicit and explicit connections. Implicit connections relate to the links between parameters influenced by task-specific loss during backpropagation, whereas explicit connections are gauged by the magnitude of parameters. Based on these, we present a new method named connection strength-based optimization for multi-task learning. Our optimization process consists of two phases. The first phase learns the task priority within the network, while the second phase modifies the gradients while upholding this priority. This ultimately leads to finding new Pareto optimal solutions for multiple tasks. Through extensive experiments with different loss scaling techniques, we show that our approach greatly enhances multi-task performance in comparison to earlier gradient manipulation methods.

## 1 INTRODUCTION

Multi-task learning (MTL) is a learning paradigm that learns multiple different tasks in a single model (Caruana, 1997). Compared to learning tasks individually, MTL can effectively reduce the number of parameters, leading to less memory usage and computation with a higher convergence rate. Furthermore, it leverages multiple tasks as an inductive bias, enabling the learning of generalized features while reducing overfitting. Complex systems such as robot vision and autonomous driving require the ability to perform multiple tasks within a single system. Thus, MTL can be a first step in finding general architecture for various domains including computer vision.

A primary goal of MTL is minimizing *negative transfer* (Crawshaw, 2020) and finding *Pareto-optimal solutions* (Sener & Koltun, 2018) for multiple tasks. Negative transfer is a phenomenon where the learning of one task adversely affects the performance of other tasks. Since each task has its own objective, this can potentially result in a trade-off among tasks. A condition in which enhancing one task is not possible without detriment to another is called *Pareto optimality*. A commonly understood cause of this trade-off is *conflicting gradients* (Yu et al., 2020) that arise during the optimization process. When the gradients of two tasks move in opposing directions, the task with larger magnitudes dominates the other, disrupting the search for new Pareto-optimal solutions. What makes more complicating the situation is unbalanced loss scales across tasks. The way we weigh task losses is crucial for multi-task performance. When there is a significant disparity in the magnitudes of losses, the task with a larger loss would dominate the entire network. Hence, the optimal strategy for MTL should efficiently handle conflicting gradients across different loss scales.

Previous studies address negative transfer by manipulating gradients or balancing tasks' losses. Solutions for handling conflicting gradients are explored in (Sener & Koltun, 2018; Yu et al., 2020; Liu et al., 2021). These approaches aim to align conflicting gradients towards a cohesive direction within a shared network space. However, these techniques are not effective at preventing negative

transfer, as they don't pinpoint which shared parameters are crucial for the tasks. This results in sub-optimal Pareto solutions for MTL, leading to pool multi-task performance. Balancing task losses is a strategy that can be applied independently from gradient manipulation methods. It includes scaling the loss according to homoscedastic uncertainty (Kendall et al., 2018), or dynamically finding loss weights by considering the rate at which the loss decreases (Liu et al., 2019).

In this paper, we propose the concept of *task priority* to address negative transfer in MTL and suggest *connection strength* as a quantifiable measure for this purpose. The task priority is defined over shared parameters by comparing the influence of each task's gradient on the overall multi-task loss. This reveals the relative importance of shared parameters to various tasks. To learn and conserve the task priority throughout the optimization process, we propose two different connections. *Implicit connections* denotes the link between shared and task-specific parameters during the backpropagation of each task-specific loss. Whereas, *explicit connections* refers to connections that can be quantified by measuring the scale of parameters. Based on the types of connection, we apply two different optimization phases. The goal of the first phase is to find a new Pareto-optimal solution for multiple tasks by learning task priority with implicit connections. The second phase is to conserve task priority learned from varying loss scales by using explicit connections. Our method outperforms previous optimization techniques that relied on gradient manipulation, consistently discovering new Pareto optimal solutions for various tasks, thereby improving multi-task performance.

## 2 RELATED WORK

**Optimization for MTL** aims to mitigate negative transfer between tasks. Some of them directly modify gradients to address task conflicts. MGDA (Désidéri, 2012; Sener & Koltun, 2018) views MTL as a multi-objective problem and minimizes the norm point in the convex hull to find a Pareto optimal set. PCGrad (Yu et al., 2020) introduces the concept of conflicting gradients and employs gradient projection to handle them. CAGrad (Liu et al., 2021) minimizes the multiple loss functions and regularizes the trajectory by leveraging the worst local improvement of individual tasks. Recon (Guangyuan et al., 2022) uses an approach similar to Neural Architecture Search (NAS) to address conflicting gradients. Some approaches use normalized gradients (Chen et al., 2018) to prevent spillover of tasks or assign stochasticity on the network's parameter based on the level of consistency in the sign of gradients (Chen et al., 2020). RotoGrad (Javaloy & Valera, 2021) rotates the feature space of the network to narrow the gap between tasks. Unlike earlier methods which guided gradients to converge towards an intermediate direction (as illustrated in Fig. 1a), our approach identifies task priority among shared parameters to update gradients, leading to Pareto-optimal solutions.

**Scaling task-specific loss** largely influences multi-task performance since the task with a significant loss would dominate the whole training process and cause severe task interference. To address the task unbalancing problem in the training, some approaches re-weight the multi-task loss by measuring homoscedastic uncertainty (Kendall et al., 2018), prioritizing tasks based on task difficulty (Guo et al., 2018), or balancing multi-task loss dynamically by considering the descending rate of loss (Liu et al., 2019). We perform extensive experiments involving different loss-scaling methods to demonstrate the robustness of our approach across various loss-weighting scenarios.

**MTL architectures** can be classified depending on the extent of network sharing across tasks. The shared trunk consists of a shared encoder followed by an individual decoder for each task (Dai et al., 2016; Ma et al., 2018; Simonyan & Zisserman, 2014; Zhang et al., 2014). Multi-modal distillation methods (Eigen & Fergus, 2015a; Xu et al., 2018; Vandenhende et al., 2020; Zhang et al., 2019) have been proposed, which can be used at the end of the shared trunk for distillation to propagate task information effectively. On the other hand, cross-talk architecture uses separate networks for each task and allows parallel information flow between layers (Gao et al., 2019). Our optimization approach can be applied to any model to mitigate task conflicts and enhance multi-task performance.

## 3 PRELIMINARIES

### 3.1 PROBLEM DEFINITION FOR MULTI-TASK LEARNING

In multi-task learning (MTL), the network learns a set of tasks $\mathcal{T} = \{\tau_1, \tau_2, ..., \tau_\mathcal{K}\}$ jointly. Each task $\tau_i$ has its own loss function $\mathcal{L}_i(\Theta)$ where $\Theta$ is the parameter of the network. The network

parameter $\Theta$ can be classified into $\Theta = \{\Theta_s, \Theta_1, \Theta_2, ..., \Theta_{\mathcal{K}}\}$ where $\Theta_s$ is shared parameter across all tasks and $\Theta_i$ is task-specific parameters devoted to task $\tau_i$. Then, the objective function of multi-task learning is to minimize the weighted sum of all tasks' losses:

$$\Theta^* = \arg\min_{\Theta} \sum_{i=1}^{\mathcal{K}} w_i \mathcal{L}_i(\Theta_s, \Theta_i) \tag{1}$$

The performance in multi-task scenarios is affected by the weighting $w_i$ of the task-specific loss $\mathcal{L}_i$.

### 3.2 Prior Approach for Multi-Task Optimization

From an optimization perspective, MTL seeks Pareto optimal solutions for multiple tasks.

**Definition 1** (Pareto optimality). *For a given network parameter $\Theta$, if we get $\Theta_{new}$ which satisfies $\mathcal{L}_i(\Theta) \geq \mathcal{L}_i(\Theta_{new})$ for all tasks $i = 1, 2, ..., \mathcal{K}$, the situation is termed a Pareto improvement. In this context, $\Theta_{new}$ is said to dominate $\Theta$. A parameter $\Theta^*$ is Pareto-optimal if no further Pareto improvements are possible. A set of Pareto optimal solutions is called a Pareto set or Pareto frontier.*

Earlier research (Sener & Koltun, 2018) interprets multi-task learning in the context of multi-objective optimization, aiming for Pareto optimality. We can empirically validate this through an analyzing the training loss and multi-task performance (Liu et al., 2021). To find Pareto optimality, some emphasize the conflicting gradients problem (Yu et al., 2020) during the optimization process.

**Definition 2** (Conflicting gradients). *Conflicting gradients are defined in the shared space of the network. Denote the gradient of task $\tau_i$ with respect to the shared parameters $\Theta_s$ as $g_i = \nabla_{\Theta_s} \mathcal{L}_i(\Theta_s, \Theta_i)$. And $g_i$ and $g_j$ are gradients of a pair of tasks $\tau_i$ and $\tau_j$ where $i \neq j$. If $g_i \cdot g_j \leq 0$, then the two gradients are called conflicting gradients.*

Previous approaches (Sener & Koltun, 2018; Yu et al., 2020; Liu et al., 2021) address the issue of conflicting gradients in shared parameters $\Theta_s$ by aligning the gradients in a consistent direction as shown in Fig. 1a. Nonetheless, they face challenges in minimizing negative transfer, as they cannot discern which parameters in $\Theta_s$ are most important to tasks. We refer to the relative importance of a task in the shared parameter as task priority. Previous studies aligned gradients without taking into account task priority, inadvertently resulting in negative transfer and reduced multi-task performance. In contrast, we introduce the notion of connection strength to determine task priority in the shared space and propose new gradient update rules based on this priority.

## 4 Method

In this section, we introduce the concept of task priority to minimize negative transfer between tasks. We utilize two distinct forms of connection to use the task priority. Following that, we propose a novel optimization method for MTL termed connection strength-based optimization. Our approach breaks down the optimization process into two phases as shown in Fig. 1b. In Phase 1, we focus on instructing the network to catch task-specific details by learning task priority. In Phase 2, task priority within the shared parameters is determined and project gradients to preserve the priority.

### 4.1 Motivation: Task priority

Using the notation given in Section 3, we propose a straightforward and intuitive analysis of our approach. Before diving deeper, we first introduce the definition of task priority.

**Definition 3** (Task priority). *Assume that the task losses $\mathcal{L}_i$ for $i = 1, 2, ..., \mathcal{K}$ are differentiable. Consider $\mathcal{X}^t$ as the input data at time $t$. We initiate with shared parameters $\Theta_s^t$ and task-specific parameters $\Theta_i^t$ with sufficiently small learning rate $\eta > 0$. A subset of shared parameters at time $t$ is denoted as $\theta^t$, such that $\theta^t \subset \Theta_s^t$. For any task $\tau_i \in \mathcal{T}$, the task's gradient for $\theta^t$ is as follows:*

$$g_i = \nabla_{\theta^t} \mathcal{L}_i(\mathcal{X}^t, \tilde{\Theta}_s^t, \theta^t, \Theta_i^t) \tag{2}$$

*where $\tilde{\Theta}_s^t$ represents the parameters that are part of $\Theta_s^t$ but not in $\theta^t$. For two distinct tasks $\tau_m, \tau_n \in \mathcal{T}$, if $\tau_m$ holds priority over $\tau_n$ in $\theta^t$, then the following inequality holds:*

$$\sum_{i=1}^{\mathcal{K}} w_i \mathcal{L}_i(\tilde{\Theta}_s^t, \theta^t - \eta \cdot g_m, \Theta_i^t) \leq \sum_{i=1}^{\mathcal{K}} w_i \mathcal{L}_i(\tilde{\Theta}_s^t, \theta^t - \eta \cdot g_n, \Theta_i^t) \tag{3}$$

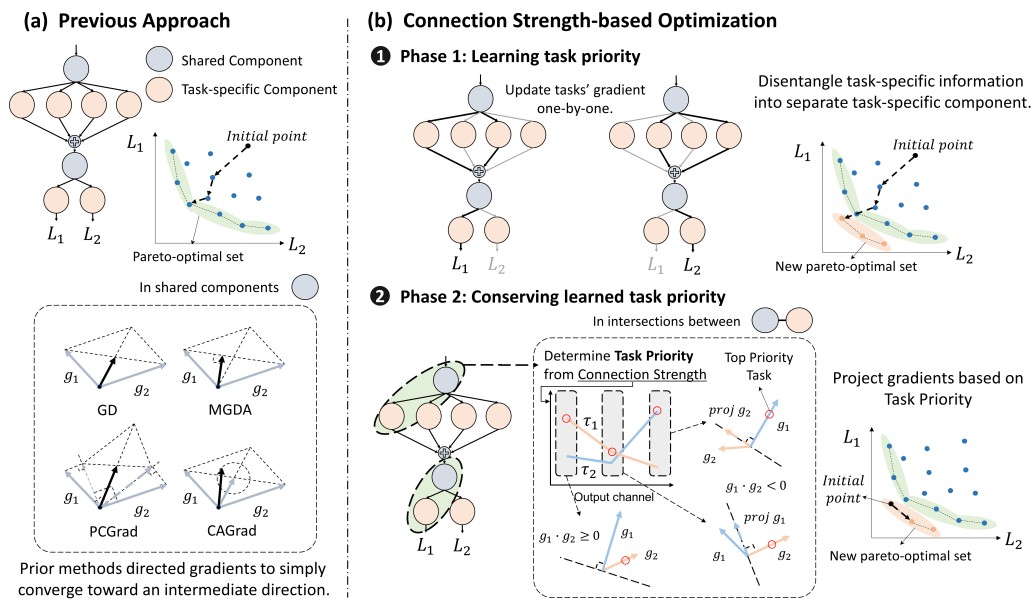

Figure 1: Overview of our connection strength-based optimization. (a) Previous methods modify gradients in shared parameters to converge toward an intermediate direction without considering the task priority, which leads to sub-optimal Pareto solutions. (b) Our method divides the optimization process into two distinct phases. In Phase 1, task priority is learned through implicit connections, leading to the identification of a new Pareto optimal solution. In Phase 2, task priority is gauged using explicit connections between shared and task-specific parameters. Subsequently, gradients are aligned with the direction of the highest-priority task's gradients. This phase ensures that priorities established in Phase 1 are maintained, thus reducing potential negative transfer.

Our motivation is to divide shared parameters $\Theta_s$ into subsets $\{\theta_{s,1}, \theta_{s,2}, ..., \theta_{s,\mathcal{K}}\}$ based on task priority. Specifically, $\theta_{s,i}$ represents a set of parameters that have a greater influence on task $\tau_i$ compared to other tasks. From the existence of the task priority, we can derive the following theorem.

**Theorem 1.** *Updating gradients based on task priority for shared parameters $\Theta_s$ (update $g_i$ for each $\theta_{s,i}$) results in a smaller multi-task loss $\sum_{i=1}^{\mathcal{K}} w_i \mathcal{L}_i$ compared to updating the weighted summation of task-specific gradients $\sum_{i=1}^{\mathcal{K}} \nabla w_i \mathcal{L}_i$ which does not take task priority into account.*

The theorem suggests that by identifying the task priority within the shared parameter $\Theta_s$, we can further expand the known Pareto frontier compared to neglecting that priority. A detailed proof is provided in Appendix A. However, identifying task priority in real-world scenarios is highly computationally demanding. This is because it requires evaluating priorities for each subset of the parameter $\Theta_s$ through pairwise comparisons among multiple tasks. Instead, we prioritize tasks based on connection strength for practical purposes.

### 4.2 CONNECTION STRENGTH

The idea of connection strength initially emerged in the field of network compression by pruning nodes in expansive convolutional neural networks (Saxena & Verbeek, 2016). This notion stems from the intuition that larger parameters have a greater influence on the model's output. Numerous studies (Han et al., 2015; Guo et al., 2016; Li et al., 2016; He et al., 2018; Yu et al., 2018; He et al., 2019; Lin et al., 2021) have reinforced this hypothesis. In our study, we re-interpret this intuition for MTL to determine task priority in shared parameters of the network.

Before we dive in, we divide network connections into two types: implicit and explicit, depending on whether their strength can be quantified or not. Conventionally, connection strength in a network refers to the connectivity between nodes, quantified by the magnitude of parameters. However, we introduce a distinct type of connection that is influenced by task-specific loss. In the context of MTL, where each task has its own distinct objective function, diverse connections are formed during the

backpropagation. Such connections are implicitly determined by the specific loss associated with each task, leading us to term them *implicit connections*. In MTL, each connection is defined for each task. A set of shared and task-specific parameters, $\Theta_s$ and $\Theta_i$ establishes a unique connection.

Conversely, *explicit connections* can be measured by the scale of parameters, mirroring the conventional notion. In this instance, we employ task-specific batch normalization to determine the task priority of the output channel of the shared convolutional layer. To establish an explicit connection, we initiate with a convolutional layer where the input is represented as $x \in \mathbf{R}^{N_I \times H \times W}$ and the weight is denoted by $W \in \mathbf{R}^{N_O \times N_I \times K \times K}$. Here, $N_I$ stands for the number of input channels, $N_O$ for the number of output channels, and $K$ indicates the kernel size. Suppose we have output channel set $\mathcal{C}^{out} = \{c_p^{out}\}_{p=1}^{N_O}$ and input channel set $\mathcal{C}^{in} = \{c_q^{in}\}_{q=1}^{N_I}$. For any given pair of output and input channels $c_p^{out} \in \mathcal{C}^{out}, c_q^{in} \in \mathcal{C}^{in}$, the connection strength $s_{p,q}$ is defined as:

$$s_{p,q} = \frac{1}{K^2} \sum_{m=0}^{K-1} \sum_{n=0}^{K-1} W(c_p^{out}, c_q^{in}, m, n)^2 \tag{4}$$

The variables $m$ and $n$ correspond to the indices of the convolutional kernel. We explore the convolutional layer followed by task-specific batch normalization, which plays a key role in determining task priority for each output channel. We revisit the equation for batch normalization with input $y$ and output $z$ of batch normalization (Ioffe & Szegedy, 2015):

$$z = \frac{\gamma}{\sqrt{Var[y] + \epsilon}} \cdot y + (\beta - \frac{\gamma E[y]}{\sqrt{Var[y] + \epsilon}}) \tag{5}$$

The coefficient of $y$ has a direct correlation with the kernel's relevance to the task since it directly modulates the output $y$. Therefore, for task $\tau_i$, we re-conceptualize the connection strength at the intersection of the convolutional layer and task-specific batch normalization in the following way:

$$S_p^{\tau_i} = \frac{\gamma_{\tau_i,p}^2}{Var[y]_p + \epsilon} \cdot \sum_{q=1}^{N_I} s_{p,q} \tag{6}$$

where $\gamma_{\tau_i,p}$ is a scale factor of the task-specific batch normalization. $S_p^{\tau_i}$ measures the contribution of each output channel $c_p^{out}$ to the output of task $\tau_i$. However, it is not possible to directly compare $S_p^{\tau_i}$ across tasks because the tasks exhibit different output scales. Hence, we employ a normalized version of connection strength that takes into account the relative scale differences among tasks:

$$\hat{S}_p^{\tau_i} = \frac{S_p^{\tau_i}}{\sum_{p=1}^{N_O} S_p^{\tau_i}} \tag{7}$$

In the following optimization, we employ two connections to learn task priority and conserve it.

### 4.3 PHASE 1: OPTIMIZATION FOR LEARNING THE TASK PRIORITY

Our first approach is very simple and intuitive. Here, the notation follows Section 3.1 and Section 4.1. For simplicity, we assume all tasks' losses are equally weighted $w_1 = w_2 = ... = w_{\mathcal{K}} = 1/\mathcal{K}$. According to conventional gradient descent (GD), we have

$$\Theta_s^{t+1} = \Theta_s^t - \eta \sum_{i=1}^{\mathcal{K}} w_i \nabla_{\Theta_s^t} \mathcal{L}_i(\mathcal{X}^t, \Theta_s^t, \Theta_i^t), \quad \Theta_i^{t+1} = \Theta_i^t - \eta \nabla_{\Theta_i^t} \mathcal{L}_i(\mathcal{X}^t, \Theta_s^t, \Theta_i^t), \quad i = 1, ..., \mathcal{K} \tag{8}$$

In standard GD, the network struggles to prioritize tasks since all tasks' gradients are updated simultaneously at each step. Instead, we sequentially update each task's gradients, as outlined below:

$$\left. \begin{cases} \Theta_s^{t+i/\mathcal{K}} = \Theta_s^{t+(i-1)/\mathcal{K}} - \eta \nabla_{\Theta_s^{t+(i-1)/\mathcal{K}}} \mathcal{L}_i(\mathcal{X}^t, \Theta_s^{t+(i-1)/\mathcal{K}}, \Theta_i^t) \\ \Theta_i^{t+1} = \Theta_i^t - \eta \nabla_{\Theta_i^t} \mathcal{L}_i(\mathcal{X}^t, \Theta_s^{t+(i-1)/\mathcal{K}}, \Theta_i^t) \end{cases} \right\} \quad i = 1, ..., \mathcal{K} \tag{9}$$

The intuition behind this optimization is to let the network divide shared parameters based on task priority, represented as $\Theta_s = \{\theta_{s,1}, \theta_{s,2}, ..., \theta_{s,\mathcal{K}}\}$. After the initial gradient descent step modifies both $\Theta_s$ and $\Theta_1$, $\theta_{s,1}$ start to better align with $\tau_1$. In the second step, the network can determine whether $\theta_{s,1}$ would be beneficial for $\tau_2$ based on the implicit connection mentioned in Section 4.2. Throughout this process, task priorities are learned by updating the task's loss in turn. Recognizing task priority in this manner effectively enables the tasks to parse out task-specific information.

---

**Algorithm 1:** Connection Strength-based Optimization for Multi-task Learning

---

**Data:** output channel set $\{c_p^{out}\}_{p=1}^{N_O}$, task set $\{\tau_i\}_{i=1}^{\mathcal{K}}$, loss function set $\{\mathcal{L}_i\}_{i=1}^{\mathcal{K}}$, channel group $\{CG_i\}_{i=1}^{\mathcal{K}}$, number of epochs $E$, current epoch $e$

1   Randomly choose $p \sim U(0,1)$
    // Phase 1: Optimization for learning the task priority
2   **if** $p \geq e/E$ **then**
3     **for** $i \leftarrow 1$ to $\mathcal{K}$ **do**
4       **update:** $g_i \leftarrow \nabla_\theta L_i$          // Update task's gradients one-by-one
    // Phase 2: Optimization for conserving the task priority
5   **else**
6     **Initialize** all $CG_i$ as empty set $\{\ \}$ in the shared convolutional layer
7     **for** $p \leftarrow 1$ to $N_O$ **do**
8       $\nu = \arg\max_i \hat{S}_p^{\tau_i}$          // Determine the task priority
9       $CG_\nu = CG_\nu + \{c_p^{out}\}$     // Classify channel with top priority task
10     **for** $i \leftarrow 1$ to $\mathcal{K}$ **do**
11       Let $\{G_{i,1}, G_{i,2}, ..., G_{i,\mathcal{K}}\}$ are gradients of $CG_i$
12       **for** $j \leftarrow 1$ to $\mathcal{K}$ and $i \neq j$ **do**
13         **if** $G_{i,i} \cdot G_{i,j} < 0$ **then**
14           $G_{i,j} = G_{i,j} - \frac{G_{i,i} \cdot G_{i,j}}{||G_{i,i}||^2} \cdot G_{i,i}$    // Project gradients with priorities
15     **update:** $g_{final} = \sum_{i=1}^{\mathcal{K}} g_i$         // Update modified gradients

---

### 4.4   PHASE 2: OPTIMIZATION FOR CONSERVING THE TASK PRIORITY

Due to negative transfer between tasks, task losses fluctuate during training, resulting in variations in multi-task performance. Therefore, we introduce a secondary optimization phase to update gradients preserving task priority. For this phase, we employ the explicit connection defined in Eq. (7). Owing to its invariant nature regarding loss scale, tasks can be prioritized regardless of their loss scale. The task priority for the channel $c_p^{out}$ is determined by evaluating the connection strength as follows:

$$\nu = \arg\max_i \hat{S}_p^{\tau_i} \tag{10}$$

After determining the priority of tasks in each output channel, the gradient vector of each task is aligned with the gradient of the highest-priority task. In detail, we categorize output channel $\{c_p^{out}\}_{p=1}^{N_O}$ into channel groups $\{CG_i\}_{i=1}^{\mathcal{K}}$ based on their primary task, denoted as $\tau_\nu$. The parameter of each channel group $CG_i$ corresponds to $\theta_{s,i}$ in $\Theta_s = \{\theta_{s,1}, \theta_{s,2}, ..., \theta_{s,\mathcal{K}}\}$. Let $\{G_{i,1}, G_{i,2}, ..., G_{i,\mathcal{K}}\}$ are task-specific gradients of $CG_i$. Then $G_{i,i}$ acts as the reference vector for identifying conflicting gradients. When another gradient vector $G_{i,j}$, where $i \neq j$, clashes with $G_{i,i}$, we adjust $G_{i,j}$ to lie on the perpendicular plane of the reference vector $G_{i,i}$ to minimize negative transfer. After projecting gradients based on task priority, the sum of them is finally updated.

In the final step, we blend two optimization stages by picking a number $p$ from a uniform distribution spanning from 0 to 1. We define $E$ as the total number of epochs and $e$ as the current epoch. The choice of optimization for that epoch hinges on whether $p$ exceeds $e/E$. As we approach the end of the training, the probability of selecting Phase 2 increases. This is to preserve the task priority learned in Phase 1 while updating the gradient in Phase 2. A detailed view of the optimization process is provided in Algorithm 1. The reason for mixing two phases instead of completely separating them is that the speed of learning task priority varies depending on the position within the network.

Previous studies (Sener & Koltun, 2018; Liu et al., 2021; Yu et al., 2020) deal with conflicting gradients by adjusting them to align in the same direction. These studies attempt to find an intermediate point among gradient vectors, which often leads to negative transfer due to the influence of the dominant task. In comparison, our approach facilitates the network's understanding of which shared parameter holds greater significance for a given task, thereby minimizing negative transfer more efficiently. The key distinction between earlier methods and ours is the inclusion of task priority.

Table 1: The experimental results of different multi-task learning optimization methods on NYUD-v2 with HRNet-18. The weights of tasks are manually tuned. Experiments are repeated over 3 random seeds and average values are presented.

| Tasks | Depth | | SemSeg | | | Surface Normal | | | | | |
|---|---|---|---|---|---|---|---|---|---|---|---|
| | Distance (Lower Better) | | (%) (Higher Better) | | | Angle Distance (Lower Better) | | Within t degree (%) (Higher Better) | | | MTP |
| Method | rmse | abs_rel | mIoU | PAcc | mAcc | mean | median | 11.25 | 22.5 | 30 | $\triangle_m \uparrow (\%)$ |
| Independent | 0.667 | 0.186 | 33.18 | 65.04 | 45.07 | 20.75 | 14.04 | 41.32 | 68.26 | 78.04 | + 0.00 |
| GD | 0.594 | 0.150 | 38.67 | 69.16 | 51.12 | 20.52 | 13.46 | 42.63 | 69.00 | 78.42 | + 9.53 |
| MGDA | 0.603 | 0.159 | 38.89 | 69.39 | 51.53 | 20.58 | 13.56 | 42.28 | 68.79 | 78.33 | + 9.21 |
| PCGrad | 0.596 | 0.149 | 38.61 | 69.30 | 51.51 | 20.50 | 13.54 | 42.56 | 69.14 | 78.55 | + 9.40 |
| CAGrad | 0.595 | 0.153 | 38.80 | 68.95 | 50.78 | 20.38 | 13.53 | 42.89 | 69.33 | 78.71 | + 9.84 |
| Ours | **0.565** | **0.148** | **41.10** | **70.37** | **53.74** | **19.54** | **12.45** | **46.11** | **71.54** | **80.12** | **+ 15.00** |

## 5 EXPERIMENTS

### 5.1 EXPERIMENTAL SETUP

**Datasets.** Our method is evaluated on three multi-task datasets: NYUD-v2 (Silberman et al., 2012), PASCAL-Context (Mottaghi et al., 2014), and Cityscapes (Cordts et al., 2016). These datasets contain different kinds of vision tasks. NYUD-v2 contains 4 vision tasks: Our evaluation is based on depth estimation, semantic segmentation, and surface normal prediction, with edge detection as an auxiliary task. PASCAL-Context contains 5 tasks: We evaluate semantic segmentation, human parts estimation, saliency estimation, and surface normal prediction, with edge detection as an auxiliary task. Cityscapes contains 2 tasks: We use semantic segmentation and depth estimation.

**Baselines.** We conduct extensive experiments with the following baselines: 1) single-task learning: training each task separately; 2) GD: simply updating all tasks' gradients jointly without any manipulation; 3) multi-task optimization methods with gradient manipulation: MGDA (Sener & Koltun, 2018), PCGrad (Yu et al., 2020), CAGrad (Liu et al., 2021); 3) loss scaling methods: We consider 4 types of loss weighting where two of them are fixed during training and the other two use dynamically varying weights. Static setting includes equal loss: all tasks are weighted equally; manually tuned loss: all tasks are weighted manually following works in (Xu et al., 2018; Vandenhende et al., 2020). Dynamic setting includes uncertainty-based approach (Kendall et al., 2018): tasks' weights are determined dynamically based on homoscedastic uncertainty; DWA (Liu et al., 2019): tasks' losses are determined considering the descending rate of loss to determine tasks' weight dynamically. 4) Architecture design methods including NAS-like approaches: Cross-Stitch (Misra et al., 2016) architecture based on SegNet (Badrinarayanan et al., 2017); Recon (Guangyuan et al., 2022): turn shared layers into task-specific layers when conflicting gradients are detected. All experiments are conducted 3 times with different random seeds for a fair comparison.

**Evaluation Metrics.** To evaluate the multi-task performance (MTP), we utilized the metric proposed in (Maninis et al., 2019). It measures the per-task performance by averaging it with respect to the single-task baseline b, as shown in $\triangle_m = (1/T) \sum_{i=1}^{T} (-1)^{l_i} (M_{m,i} - M_{b,i})/M_{b,i}$ where $l_i = 1$ if a lower value of measure $M_i$ means better performance for task $i$, and 0 otherwise. We measured the single-task performance of each task $i$ with the same backbone as baseline $b$. To evaluate the performance of tasks, we employed widely used metrics. More details are provided in Appendix C.

### 5.2 EXPERIMENTAL RESULTS

**Our method achieves the largest improvements in multi-task performance.** The main results on NYUD-v2, PASCAL-Context are presented in Table 1 and Table 2 respectively. For a fair comparison, we compare various optimization methods on exactly the same architecture with identical task-specific layers. Tasks' losses are tuned manually following the setting in (Xu et al., 2018; Vandenhende et al., 2020). Compared to previous methods, our approach shows better performance on most tasks and datasets. It proves our method tends to induce less task interference.

Table 2: The experimental results of different multi-task learning optimization methods on PASCAL-Context with HRNet-18. The weights of tasks are manually tuned. Experiments are repeated over 3 random seeds and average values are presented.

| Tasks | SemSeg | | PartSeg | Saliency | | Surface Normal | | | | | |
|---|---|---|---|---|---|---|---|---|---|---|---|
| Method | (Higher Better) | | (Higher Better) | (Higher Better) | | Angle Distance (Lower Better) | | Within t degree (%) (Higher Better) | | | MTP |
| | mIoU | PAcc | mIoU | mIoU | maxF | mean | median | 11.25 | 22.5 | 30 | $\triangle_m \uparrow (\%)$ |
| Independent | 60.30 | 89.88 | 60.56 | 67.05 | 78.98 | 14.76 | 11.92 | 47.61 | 81.02 | 90.65 | + 0.00 |
| GD | 62.17 | 90.27 | 61.15 | 67.99 | 79.60 | 14.70 | 11.81 | 47.55 | 80.97 | 90.56 | + 1.47 |
| MGDA | 61.75 | 89.98 | 61.69 | 67.32 | 78.98 | 14.77 | 12.22 | 47.02 | 80.91 | 90.14 | + 1.15 |
| PCGrad | 62.47 | 90.57 | 61.46 | 67.86 | 79.38 | 14.59 | 11.77 | 47.72 | 81.28 | 90.81 | + 1.86 |
| CAGrad | 62.22 | 90.01 | 61.89 | 67.46 | 79.12 | 14.97 | 12.10 | 47.23 | 80.54 | 90.30 | + 1.14 |
| Ours | **63.86** | **90.65** | **63.05** | **68.30** | **79.26** | **14.33** | **11.45** | **49.08** | **81.86** | **91.05** | **+ 3.70** |

**Proposed optimization works robustly on various loss scaling methods.** To prove the generality of our optimization method, we conduct extensive experiments on NYUD-v2 as shown in Tables 1 and 5 to 7 and PASCAL-Context as shown in Tables 2 and 12 to 14. In almost all types of loss scaling, our method shows the best multi-task performance. Unlike conventional approaches where the effectiveness of optimization varies depending on the loss scaling method, ours can be applied to various types of loss weighting and shows robust results.

**Our method can be applied to various types of network architecture.** We use MTI-Net (Vandenhende et al., 2020) with HRNet-18 (Wang et al., 2020) and ResNet-18 (He et al., 2016) on NYUD-v2 and PASCAL-Context. HRNet-18 and ResNet-18 are pre-trained on ImageNet (Krizhevsky et al., 2017). On the other hand, we use SegNet (Badrinarayanan et al., 2017) for Cityscapes from scratch following the experiments setting in (Liu et al., 2021; Guangyuan et al., 2022). Our optimization shows robustly better performance with different neural network architectures. The results with ResNet-18 are also experimented with various loss scaling as shown in Tables 8 to 11.

**Results are compatible with various architectures with fewer parameters.** In Table 3, we evaluate our methods in different aspects by considering the various types of architecture. In the table, we include the results of Recon (Guangyuan et al., 2022) to show our method can mitigate negative transfer between tasks more parameter efficiently. Compared to Cross-Stitch (Misra et al., 2016) and RotoGrad (Javaloy & Valera, 2021), ours show better multi-task performance with fewer parameters. Compared to Recon, our method is more parameter efficient as it increases the number of parameters by about 0.05% with the use of task-specific batch normalization. Our method shows comparable performance on Cityscapes with fewer parameters.

**Our method finds new Pareto optimal solutions for multiple tasks.** The final task-specific loss and their average are shown in Fig. 2 for NYUD-v2 and PASCAL-Context. We compare our method with previous gradient manipulation techniques and repeat the experiments over 3 random seeds. For both NYUD-v2 and PASCAL-Context, ours show the lowest average training loss. When comparing each task individually, ours still shows the lowest final loss on every task. This provides concrete proof that our method leads to the expansion of the known Pareto frontier of previous approaches.

## 5.3 ABLATION STUDY

**Phase 1 learns task priority to find Pareto-optimal solutions.** We perform ablation studies on each stage of optimization as shown in Table 4. When solely utilizing phase 2, its performance has no big difference from the previous optimization techniques. However, when the first phase was used, the lowest averaged multi-task loss was achieved. Additionally, we show the correlation of loss trends in Fig. 3. The closer the value is to 1, the more it means that the loss of the task pair decreases together. In the initial stages of optimization, phase 1 appears to align the loss more effectively than solely relying on phase 2. This shows that phase 1 aids the network in differentiating task-specific details, leading to the identification of optimal Pareto solutions.

**During Phase 2, the task's priority is likely to be maintained.** We visualized the percentage of top priority tasks in Fig. 4 by measuring connection strength Eq. (7). The figure illustrates how much of the output channels in the shared convolutional layer each task has priority over. We compared when we used only Phase 1 and when we used both Phase 1 and Phase 2. We found Phase 2 at the latter

Table 3: The comparison of multi-task performance on Cityscapes. Ours demonstrate competitive results without any significant addition to the network's parameters.

| Method | Segmentation (Higher Better) | | Depth (Lower Better) | | $\triangle_m \uparrow(\%)$ | #P. |
|---|---|---|---|---|---|---|
| | mIoU | Pix Acc | Abs Err | Rel Err | | |
| Single-task | 74.36 | 93.22 | 0.0128 | 29.98 | - 79.04 | 190.59 |
| Cross-Stitch | 74.05 | 93.17 | 0.0162 | 116.66 | - 79.04 | 190.59 |
| RotoGrad | 73.38 | 92.97 | 0.0147 | 82.31 | - 47.81 | 103.43 |
| GD | 74.13 | 93.13 | 0.0166 | 116.00 | - 79.32 | 95.43 |
| w/ Recon | 71.17 | 93.21 | 0.0136 | 43.18 | - 12.63 | 108.44 |
| MGDA | 70.74 | 92.19 | 0.0130 | 47.09 | - 16.22 | 95.43 |
| w/ Recon | 71.01 | 92.17 | 0.0129 | **33.41** | **- 4.46** | 108.44 |
| Graddrop | 74.08 | 93.08 | 0.0173 | 115.79 | - 80.48 | 95.43 |
| w/ Recon | 74.17 | 93.11 | 0.0134 | 41.37 | - 10.69 | 108.44 |
| PCGrad | 73.98 | 93.08 | 0.02 | 114.50 | - 78.39 | 95.43 |
| w/ Recon | 74.18 | 93.14 | 0.0136 | 46.02 | - 14.92 | 108.44 |
| CAGrad | 73.81 | 93.02 | 0.0153 | 88.29 | - 53.81 | 95.43 |
| w/ Recon | 74.22 | 93.10 | 0.0130 | 38.27 | - 7.38 | 108.44 |
| Ours | **74.75** | **93.39** | **0.0125** | 41.60 | - 10.08 | 95.48 |

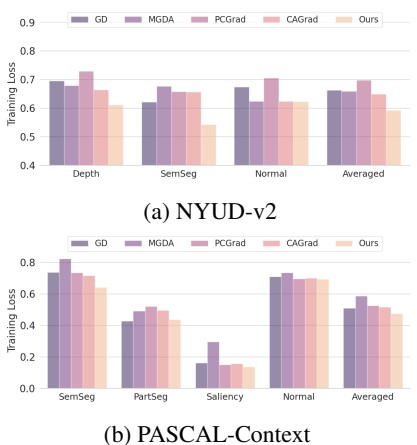

(a) NYUD-v2

(b) PASCAL-Context

Figure 2: The comparison of training losses on the NYUDv2 and PASCAL-Context. Ours find a new Pareto optimal solution for multiple tasks.

Table 4: Comparison of multi-task performance using each phase individually, sequentially, and by the proposed mixing method on NYUD-v2.

| Phase | | Depth | Seg | Norm | MTP | Averaged |
|---|---|---|---|---|---|---|
| 1 | 2 | rmse | mIoU | mean | $\triangle_m \uparrow$ | Loss |
| ✓ | | 0.581 | 40.36 | 19.55 | + 13.44 | **0.5396** |
| | ✓ | 0.597 | 39.23 | 20.39 | + 10.32 | 0.6519 |
| ✓$_{seq}$ | ✓$_{seq}$ | 0.574 | 40.38 | 19.56 | + 13.79 | 0.5788 |
| ✓$_{mix}$ | ✓$_{mix}$ | **0.565** | **41.10** | **19.54** | **+ 15.50** | 0.5942 |

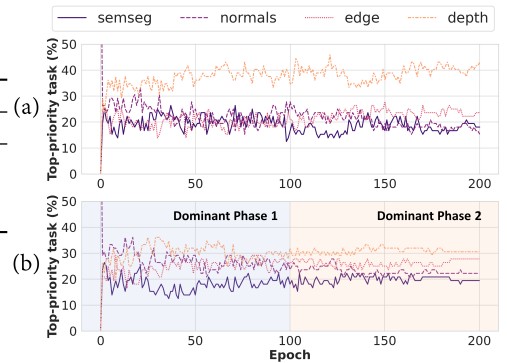

Figure 3: Correlation of loss trends across tasks during the epochs. a) Phase 1, b) Phase 2.

(a)

(b)

Figure 4: Visualization of the percentage of top-priority tasks over training epoch. a) Phase 1, b) Mixing Phase 1 and Phase 2

half of the optimization has an effect on conserving learned task priority. This method of priority allocation prevents a specific task from exerting a dominant influence over the entire network.

**Mixing two phases shows higher performance than using each phase separately.** In Table 4, using only Phase 1 results in a lower multi-task loss than when mixing the two phases. Nonetheless, combining both phases enhances multi-task performance. This improvement can be attributed to the normalized connection strength (refer to Eq. (7)), which ensures that no single task dominates the entire network during Phase 2. When the two phases are applied sequentially, performance declines compared to our mixing strategy. The reason for this performance degradation seems to be the application of Phase 1 at the later stages of Optimization. This continuously alters the established task priority, which in turn disrupts the gradient's proper updating based on the learned priority.

## 6  CONCLUSION

In this paper, we present a novel optimization technique for multi-task learning named connection strength-based optimization. By recognizing task priority within shared network parameters and measuring it using connection strength, we pinpoint which portions of these parameters are crucial for distinct tasks. By learning and preserving this task priority during optimization, we are able to identify new Pareto optimal solutions, boosting multi-task performance. We validate the efficacy of our strategy through comprehensive experimentation and analysis.

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

## A   PROOF OF THEOREM 1

**Theorem 1.** *Updating gradients based on task priority for shared parameters $\Theta_s$ (update $g_i$ for each $\theta_{s,i}$) results in a smaller multi-task loss $\sum_{i=1}^{\mathcal{K}} w_i \mathcal{L}_i$ compared to updating the weighted summation of task-specific gradients $\sum_{i=1}^{\mathcal{K}} \nabla w_i \mathcal{L}_i$ which does not take task priority into account.*

*Proof.* We start from shared parameters $\Theta_s$ and we can divide them with task priority.

$$\Theta_s = \{\theta_{s,1}, \theta_{s,2}, ..., \theta_{s,\mathcal{K}}\} \tag{11}$$

For the sake of simplicity in our proof, we begin by focusing on a subset of shared parameters, specifically $\theta_{s,i}$, to demonstrate that accounting for task priority leads to a reduced multi-task loss compared to neglecting it. Subsequently, we will apply the same process to the remaining shared parameters to complete the proof. Let $\hat{g}_k$ be the gradient of $\theta_{s,i}^t$ for task $\tau_k$ as follows:

$$\hat{g}_k = \nabla_{\theta_{s,i}^t} \mathcal{L}_k(\mathcal{X}^t, \tilde{\Theta}_s^t, \theta_{s,i}^t, \Theta_i^t) \tag{12}$$

Previous optimization methods involving gradient manipulation update the weighted summation of task-specific gradients. Therefore, we can update $\theta_{s,i}^t$ to $\theta_{s,i}^{t+1}$ as follows:

$$g = \sum_{j=1}^{\mathcal{K}} \nabla_{\theta_{s,i}^t} w_j \mathcal{L}_j(\mathcal{X}^t, \tilde{\Theta}_s^t, \theta_{s,i}^t, \Theta_i^t) = \sum_{j=1}^{\mathcal{K}} w_j \hat{g}_j, \qquad \theta_{s,i}^{t+1} = \theta_{s,i}^t - \eta g \tag{13}$$

where $w_i$ is loss weights of $\tau_i$ and $\sum_{i=1}^{\mathcal{K}} w_i = 1$.

From the first order Taylor approximation of $\mathcal{L}_i$ for $\theta_{s,i}$, we have

$$\mathcal{L}_i(\mathcal{X}^t, \tilde{\Theta}_s^t, \theta_{s,i}^{t+1}, \Theta_i^t) = \mathcal{L}_i(\mathcal{X}^t, \tilde{\Theta}_s^t, \theta_{s,i}^t, \Theta_i^t) + (\theta_{s,i}^{t+1} - \theta_{s,i}^t)^\top \hat{g}_i + O(\eta^2) \tag{14}$$

On the other hand, when considering task priority, we can update $\theta_{s,i}^t$ to $\hat{\theta}_{s,i}^{t+1}$ using $\hat{g}_i$ as follows:

$$\hat{\theta}_{s,i}^{t+1} = \theta_{s,i}^t - \eta \hat{g}_i \tag{15}$$

From the first order Taylor approximation of $\mathcal{L}_i$ from $\theta_{s,i}^t$ to $\hat{\theta}_{s,i}^{t+1}$, we have

$$\mathcal{L}_i(\mathcal{X}^t, \tilde{\Theta}_s^t, \hat{\theta}_{s,i}^{t+1}, \Theta_i^t) = \mathcal{L}_i(\mathcal{X}^t, \tilde{\Theta}_s^t, \theta_{s,i}^t, \Theta_i^t) + (\hat{\theta}_{s,i}^{t+1} - \theta_{s,i}^t)^\top \hat{g}_i + O(\eta^2) \tag{16}$$

The difference between Eq. (14) and Eq. (16) is

$$\mathcal{L}_i(\mathcal{X}^t, \tilde{\Theta}_s^t, \theta_{s,i}^{t+1}, \Theta_i^t) - \mathcal{L}_i(\mathcal{X}^t, \tilde{\Theta}_s^t, \hat{\theta}_{s,i}^{t+1}, \Theta_i^t) = (\theta_{s,i}^{t+1} - \theta_{s,i}^t)^\top \hat{g}_i - (\hat{\theta}_{s,i}^{t+1} - \theta_{s,i}^t)^\top \hat{g}_i \tag{17}$$

$$= -\eta(g - \hat{g}_i)^\top \hat{g}_i \tag{18}$$

Similarly, for the last of losses $\mathcal{L}_j$ where $i \neq j$, we have

$$\mathcal{L}_j(\mathcal{X}^t, \tilde{\Theta}_s^t, \theta_{s,i}^{t+1}, \Theta_i^t) = \mathcal{L}_j(\mathcal{X}^t, \tilde{\Theta}_s^t, \theta_{s,i}^t, \Theta_i^t) + (\theta_{s,i}^{t+1} - \theta_{s,i}^t)^\top \hat{g}_j + O(\eta^2) \tag{19}$$

$$\mathcal{L}_j(\mathcal{X}^t, \tilde{\Theta}_s^t, \hat{\theta}_{s,i}^{t+1}, \Theta_i^t) = \mathcal{L}_j(\mathcal{X}^t, \tilde{\Theta}_s^t, \theta_{s,i}^t, \Theta_i^t) + (\hat{\theta}_{s,i}^{t+1} - \theta_{s,i}^t)^\top \hat{g}_j + O(\eta^2) \tag{20}$$

The difference between Eq. (19) and Eq. (20) is

$$\mathcal{L}_j(\mathcal{X}^t, \tilde{\Theta}_s^t, \theta_{s,i}^{t+1}, \Theta_i^t) - \mathcal{L}_j(\mathcal{X}^t, \tilde{\Theta}_s^t, \hat{\theta}_{s,i}^{t+1}, \Theta_i^t) = (\theta_{s,i}^{t+1} - \theta_{s,i}^t)^\top \hat{g}_j - (\hat{\theta}_{s,i}^{t+1} - \theta_{s,i}^t)^\top \hat{g}_j \tag{21}$$

$$= -\eta(g - \hat{g}_i)^\top \hat{g}_j \tag{22}$$

If we sum Eq. (22) over all task-specific losses $\mathcal{L}_1, \mathcal{L}_2, ..., \mathcal{L}_\mathcal{K}$, then we have

$$\sum_{k=1}^{\mathcal{K}} \mathcal{L}_k(\mathcal{X}^t, \tilde{\Theta}_s^t, \theta_{s,i}^{t+1}, \Theta_i^t) - \sum_{k=1}^{\mathcal{K}} \mathcal{L}_k(\mathcal{X}^t, \tilde{\Theta}_s^t, \hat{\theta}_{s,i}^{t+1}, \Theta_i^t) \tag{23}$$

$$= -\sum_{k=1}^{\mathcal{K}} \eta(g - \hat{g}_i)^\top \hat{g}_k \tag{24}$$

$$= -\sum_{k=1}^{\mathcal{K}} \eta(\sum_{j=1}^{\mathcal{K}} \nabla_{\theta_{s,i}^t} w_j \mathcal{L}_j(\mathcal{X}^t, \tilde{\Theta}_s^t, \theta_{s,i}^t, \Theta_i^t) - \nabla_{\theta_{s,i}^t} \mathcal{L}_i(\mathcal{X}^t, \tilde{\Theta}_s^t, \theta_{s,i}^t, \Theta_i^t))^\top \hat{g}_k \tag{25}$$

$$= -\sum_{k=1}^{\mathcal{K}} \eta \left( \sum_{j=1}^{\mathcal{K}} w_j \left( \nabla_{\theta_{s,i}^t} \mathcal{L}_j(\mathcal{X}^t, \tilde{\Theta}_s^t, \theta_{s,i}^t, \Theta_i^t) - \nabla_{\theta_{s,i}^t} \mathcal{L}_i(\mathcal{X}^t, \tilde{\Theta}_s^t, \theta_{s,i}^t, \Theta_i^t) \right) \right)^\top \hat{g}_k \tag{26}$$

$$\geq 0 \tag{27}$$

The elements within the brackets of Eq. (26) represent a pairwise comparison of the changes in loss resulting from updating the gradients of each task. Thus, the inequality of Eq. (27) holds from Definition 3 of task priority. The results indicate that taking task priority into account yields a lower multi-task loss compared to neglecting it.

Following a similar process for all shared parameters $\Theta_s = \{\theta_{s,1}, \theta_{s,2}, ..., \theta_{s,\mathcal{K}}\}$, we can conclude considering task priority leads to the expansion of the known Pareto frontier.

$\square$

## B  LOSS SCALING METHODS

In this paper, we used 4 different loss scaling methods to weigh multiple tasks' losses.

1. All tasks' losses are weighted equally.

2. The weights of tasks are tuned manually following the previous works (Xu et al., 2018; Vandenhende et al., 2020). For NYUD-v2, the weight of losses is as follows:
    Depth : SemSeg : Surface Normal : Edge = 1.0 : 1.0 : 10.0 : 50.0
 For PASCAL-Context, the weight of losses is as follows:
    Semseg : PartSeg : Saliency : Surface Normal : Edge = 1.0 : 2.0 : 5.0 : 10.0 : 50.0

3. The losses are dynamically weighted by homoscedastic uncertainty (Kendall et al., 2018).
 An uncertainty that cannot be reduced with increasing data is called Aleatoric uncertainty. Homoscedastic uncertainty is a kind of Aleatoric uncertainty that stays constant for all input data and varies between different tasks. So it is also called task-dependent uncertainty. Homoscedastic uncertainty is formulated differently depending on whether the task is a regression task or a classification task as each of them uses different output functions: A regression task uses Gaussian Likelihood, in contrast, a classification task uses softmax function. The objectives of uncertainty weighting are as follows:

$$\mathcal{L}_{Total} = \sum_{i=1}^{\mathcal{K}} \hat{\mathcal{L}}_i \quad where \quad \hat{\mathcal{L}}_i = \begin{cases} \dfrac{1}{2\sigma_1^2}\mathcal{L}_i + \log \sigma_i & \text{for regression task} \\ \dfrac{1}{\sigma_2^2}\mathcal{L}_i + \log \sigma_i & \text{for classification task} \end{cases} \tag{28}$$

4. The losses are dynamically weighted by descending rate of loss (Liu et al., 2019) which is called Dynamic Weight Average (DWA). The weight of task $w_i$ is defined as follows with DWA:

$$w_i(t) = \frac{\mathcal{K} \exp(w_i(t-1)/T)}{\sum_{i=1}^{\mathcal{K}} exp(w_i(t-1)/T)} \quad where \quad w_i(t-1) = \frac{\mathcal{L}_k(t-1)}{\mathcal{L}_k(t-2)} \tag{29}$$

where $t$ is an iteration index and $\mathcal{K}$ is the number of tasks. $T$ represents the temperature parameter governing the softness of task weighting. As $T$ increases, the tasks become likely to be weighted equally. We used $T = 2$ for our experiments following the works in (Liu et al., 2019).

# C    EXPERIMENTAL DETAILS

**Implementation details.**  To train MTI-Net (Vandenhende et al., 2020) on both NYUD-v2 and PASCAL-Context, we adopted the loss schema and augmentation strategy from PAD-Net(Xu et al., 2018) and MTI-Net(Vandenhende et al., 2020). For depth estimation, we utilized L1 loss, while the cross-entropy loss was used for semantic segmentation. To train for saliency estimation and edge detection, we employed the well-known balanced cross-entropy loss. Surface normal prediction used L1 loss. We augmented input images by randomly scaling them with a ratio from 1, 1.2, 1.5 and horizontally flipping them with a 50% probability. The network was trained for 200 epochs for NYUD-v2 and 50 epochs for PASCAL-Context using the Adam optimizer. We employed a learning rate of $10^{-4}$ with a poly learning rate decay policy. We used a weight decay of $10^{-4}$ and batch size of 8.

In contrast, for Cityscapes with SegNet (Badrinarayanan et al., 2017), we followed the experimental setting in (Liu et al., 2021; Guangyuan et al., 2022). We used L1 loss and cross-entropy loss for depth estimation and semantic segmentation, respectively. The network was trained for 200 epochs using the Adam optimizer. We employed a learning rate of $5 \times 10^{-5}$ with multi-step learning rate scheduling. We used a batch size of 8.

**Evaluation metric.**  To evaluate the performance of tasks, we employed widely used metrics. For semantic segmentation, we utilized mean Intersection over Union (mIoU), Pixel Accuracy (PAcc), and mean Accuracy (mAcc). Surface normal prediction's performance was measured by calculating the mean and median angle distances between the predicted output and ground truth. We also used the proportion of pixels within the angles of $11.25°$, $22.5°$, $30°$ to the ground truth, as suggested by (Eigen & Fergus, 2015b). To evaluate the depth estimation task, we followed the methods proposed in (Eigen et al., 2014; Liu et al., 2015; Xu et al., 2017). We used Root Mean Squared Error (RMSE), and Mean Relative Error (abs_rel). For saliency estimation and human part segmentation, we employed mean Intersection over Union (mIoU).

# D    ADDITIONAL EXPERIMENTAL RESULTS

We compare GD, MGDA (Sener & Koltun, 2018), PCGrad (Yu et al., 2020), CAGrad (Liu et al., 2021), and connection strength based optimization on 4 different multi-task loss scaling methods mentioned in Appendix B. We have summarized the experimental overview as follows.

1. NYUD-v2 with HRNet-18 on various loss scaling is evaluated in Tables 5 to 7.

2. NYUD-v2 with ResNet-18 on various loss scaling is evaluated in Tables 8 to 11.

3. PASCAL-Context with HRNet-18 on various loss scaling is evaluated in Tables 12 to 14.

## D.1    NYUD-V2 WITH HRNET-18

Table 5: The experimental results of different multi-task optimization methods on NYUD-v2 with HRNet-18. The losses of all tasks are evenly weighted. Experiments are repeated over 3 random seeds and average values are presented. $\triangle_m \uparrow(\%)$ is used to indicate the percentage improvement in multi-task performance (MTP). The best results are expressed in **bold** numbers.

| Tasks | Depth | | SemSeg | | | Surface Normal | | | | | |
|---|---|---|---|---|---|---|---|---|---|---|---|
| | Distance (Lower Better) | | (%) (Higher Better) | | | Angle Distance (Lower Better) | | Within t degree (%) (Higher Better) | | | MTP |
| Method | rmse | abs_rel | mIoU | PAcc | mAcc | mean | median | 11.25 | 22.5 | 30 | $\triangle_m \uparrow(\%)$ |
| Independent | 0.667 | 0.186 | 33.18 | 65.04 | 45.07 | 20.75 | 14.04 | 41.32 | 68.26 | 78.04 | + 0.00 |
| GD | 0.595 | 0.150 | 40.67 | 70.11 | 53.41 | 21.45 | 15.02 | 39.06 | 66.42 | 76.87 | + 10.00 |
| MGDA | 0.587 | 0.148 | 40.69 | 70.40 | 53.15 | 21.30 | 14.73 | 39.59 | 66.85 | 77.12 | + 10.66 |
| PCGrad | 0.581 | 0.155 | 40.33 | 70.44 | 52.83 | 21.23 | 14.59 | 40.01 | 67.17 | 77.31 | + 10.71 |
| CAGrad | **0.576** | 0.149 | 40.00 | 70.45 | 51.75 | 21.09 | 14.50 | 40.18 | 67.40 | 77.47 | + 10.85 |
| Ours | **0.576** | **0.143** | **41.20** | **71.03** | **53.76** | **20.42** | **13.75** | **42.20** | **69.22** | **78.88** | **+ 13.13** |

Table 6: The experimental results of different multi-task optimization methods on NYUD-v2 with HRNet-18. The losses are weighted using Dynamic Weight Average (DWA). Experiments are repeated over 3 random seeds and average values are presented. $\triangle_m \uparrow(\%)$ is used to indicate the percentage improvement in multi-task performance (MTP). The best results are expressed in **bold** numbers.

| Tasks | Depth | | SemSeg | | | Surface Normal | | | | | |
| Method | Distance (Lower Better) | | (%) (Higher Better) | | | Angle Distance (Lower Better) | | Within t degree (%) (Higher Better) | | | MTP |
| | rmse | abs_rel | mIoU | PAcc | mAcc | mean | median | 11.25 | 22.5 | 30 | $\triangle_m \uparrow(\%)$ |
| Independent | 0.667 | 0.186 | 33.18 | 65.04 | 45.07 | 20.75 | 14.04 | 41.32 | 68.26 | 78.04 | + 0.00 |
| GD | 0.592 | 0.146 | 40.86 | 70.19 | 53.01 | 21.15 | 14.52 | 40.20 | 67.36 | 77.48 | + 10.82 |
| MGDA | 0.593 | 0.147 | 40.46 | 70.10 | 52.83 | 21.30 | 14.68 | 39.73 | 66.90 | 77.16 | + 10.13 |
| PCGrad | 0.593 | 0.147 | 40.34 | 70.00 | 52.37 | 21.36 | 14.77 | 39.57 | 66.78 | 77.07 | + 9.91 |
| CAGrad | 0.576 | 0.146 | 40.52 | 70.23 | 52.73 | 21.09 | 14.59 | 40.18 | 67.40 | 77.49 | + 11.38 |
| Ours | **0.565** | **0.141** | **41.64** | **70.97** | **54.49** | **20.35** | **13.48** | **43.04** | **69.60** | **78.95** | **+ 14.24** |

Table 7: The experimental results of different multi-task optimization methods on NYUD-v2 with HRNet-18. The losses are weighted by homoscedastic uncertainty. Experiments are repeated over 3 random seeds and average values are presented. $\triangle_m \uparrow(\%)$ is used to indicate the percentage improvement in multi-task performance (MTP). The best results are expressed in **bold** numbers.

| Tasks | Depth | | SemSeg | | | Surface Normal | | | | | |
| Method | Distance (Lower Better) | | (%) (Higher Better) | | | Angle Distance (Lower Better) | | Within t degree (%) (Higher Better) | | | MTP |
| | rmse | abs_rel | mIoU | PAcc | mAcc | mean | median | 11.25 | 22.5 | 30 | $\triangle_m \uparrow(\%)$ |
| Independent | 0.667 | 0.186 | 33.18 | 65.04 | 45.07 | 20.75 | 14.04 | 41.32 | 68.26 | 78.04 | + 0.00 |
| GD | 0.589 | 0.148 | 39.93 | 70.15 | 51.99 | 21.13 | 14.46 | 40.47 | 67.28 | 77.38 | + 9.87 |
| MGDA | 0.590 | 0.148 | 39.78 | 69.77 | 51.80 | 21.24 | 14.69 | 39.78 | 66.94 | 77.22 | + 9.69 |
| PCGrad | 0.587 | 0.147 | 40.56 | 69.97 | 53.07 | 21.19 | 14.40 | 40.51 | 67.46 | 77.41 | + 10.71 |
| CAGrad | 0.583 | 0.147 | 40.23 | 70.06 | 52.74 | 21.09 | 14.47 | 40.23 | 67.48 | 77.50 | + 10.73 |
| Ours | **0.569** | **0.140** | **41.16** | **70.83** | **53.65** | **20.19** | **13.39** | **43.33** | **70.07** | **79.30** | **+ 13.81** |

## D.2 NYUD-v2 WITH RESNET-18

Table 8: The experimental results of different multi-task optimization methods on NYUD-v2 with ResNet-18. The losses of all tasks are evenly weighted. Experiments are repeated over 3 random seeds and average values are presented. $\triangle_m \uparrow(\%)$ is used to indicate the percentage improvement in multi-task performance (MTP). The best results are expressed in **bold** numbers.

| Tasks | Depth | | SemSeg | | | Surface Normal | | | | | |
| Method | Distance (Lower Better) | | (%) (Higher Better) | | | Angle Distance (Lower Better) | | Within t degree (%) (Higher Better) | | | MTP |
| | rmse | abs_rel | mIoU | PAcc | mAcc | mean | median | 11.25 | 22.5 | 30 | $\triangle_m \uparrow(\%)$ |
| Independent | 0.659 | 0.183 | 34.46 | 65.51 | 46.50 | 23.36 | 16.67 | 34.89 | 62.19 | 73.33 | + 0.00 |
| GD | 0.613 | **0.160** | 38.54 | 68.89 | 51.04 | 22.09 | 15.35 | 38.29 | 65.12 | 75.61 | + 8.09 |
| MGDA | 0.616 | 0.165 | **39.49** | 69.30 | **52.30** | 22.52 | 15.61 | 37.92 | 64.25 | 74.77 | + 8.24 |
| PCGrad | 0.618 | 0.164 | 38.76 | 69.01 | 51.12 | 22.05 | 15.28 | 38.55 | 65.36 | 75.77 | + 8.10 |
| CAGrad | 0.610 | **0.160** | 39.20 | **69.38** | 51.58 | 22.18 | 15.61 | 37.65 | 64.70 | 75.42 | + 8.75 |
| Ours | **0.601** | 0.162 | 38.30 | 68.78 | 51.01 | **21.09** | **14.31** | **40.95** | **67.57** | **77.50** | **+ 9.89** |

Table 9: The experimental results of different multi-task optimization methods on NYUD-v2 with ResNet-18. The weights of tasks are manually tuned. Experiments are repeated over 3 random seeds and average values are presented. $\triangle_m \uparrow(\%)$ is used to indicate the percentage improvement in multi-task performance (MTP). The best results are expressed in **bold** numbers.

| Tasks | Depth | | SemSeg | | | Surface Normal | | | | | |
|---|---|---|---|---|---|---|---|---|---|---|---|
| | Distance (Lower Better) | | (%) (Higher Better) | | | Angle Distance (Lower Better) | | Within t degree (%) (Higher Better) | | | MTP |
| Method | rmse | abs_rel | mIoU | PAcc | mAcc | mean | median | 11.25 | 22.5 | 30 | $\triangle_m \uparrow(\%)$ |
| Independent | 0.659 | 0.183 | 34.46 | 65.51 | 46.50 | 23.36 | 16.67 | 34.89 | 62.19 | 73.33 | + 0.00 |
| GD | 0.622 | 0.163 | 38.07 | 68.31 | 50.84 | 21.49 | 14.63 | 40.04 | 66.87 | 76.87 | + 8.03 |
| MGDA | 0.635 | 0.166 | 38.18 | 68.22 | 49.70 | 22.07 | 15.01 | 39.11 | 65.81 | 75.90 | + 6.65 |
| PCGrad | 0.617 | 0.165 | 37.80 | 67.94 | 50.00 | 21.52 | 14.53 | 40.27 | 66.91 | 76.71 | + 7.98 |
| CAGrad | 0.620 | 0.163 | 37.02 | 67.96 | 49.71 | 21.67 | 14.80 | 39.55 | 66.46 | 76.56 | + 6.86 |
| Ours | **0.600** | **0.157** | **39.00** | **69.02** | **51.11** | **20.65** | **13.77** | **42.78** | **68.97** | **78.30** | **+ 11.24** |

Table 10: The experimental results of different multi-task optimization methods on NYUD-v2 with ResNet-18. The losses are weighted using Dynamic Weight Average (DWA). Experiments are repeated over 3 random seeds and average values are presented. $\triangle_m \uparrow(\%)$ is used to indicate the percentage improvement in multi-task performance (MTP). The best results are expressed in **bold** numbers.

| Tasks | Depth | | SemSeg | | | Surface Normal | | | | | |
|---|---|---|---|---|---|---|---|---|---|---|---|
| | Distance (Lower Better) | | (%) (Higher Better) | | | Angle Distance (Lower Better) | | Within t degree (%) (Higher Better) | | | MTP |
| Method | rmse | abs_rel | mIoU | PAcc | mAcc | mean | median | 11.25 | 22.5 | 30 | $\triangle_m \uparrow(\%)$ |
| Independent | 0.659 | 0.183 | 34.46 | 65.51 | 46.50 | 23.36 | 16.67 | 34.89 | 62.19 | 73.33 | + 0.00 |
| GD | 0.607 | 0.159 | 38.65 | 68.99 | 51.72 | 22.17 | 15.52 | 38.51 | 65.11 | 75.47 | + 8.38 |
| MGDA | 0.616 | 0.165 | 39.38 | 69.18 | 51.78 | 22.53 | 15.69 | 37.68 | 64.12 | 74.67 | + 8.12 |
| PCGrad | 0.612 | 0.162 | 38.56 | 68.97 | 51.16 | 22.11 | 15.40 | 38.20 | 65.07 | 75.58 | + 8.13 |
| CAGrad | 0.609 | 0.157 | **39.40** | **69.30** | **51.84** | 22.28 | 15.68 | 37.62 | 64.46 | 75.24 | + 8.85 |
| Ours | **0.592** | **0.148** | 38.41 | 68.82 | 51.15 | **20.96** | **14.25** | **40.97** | **67.59** | **77.10** | **+ 10.63** |

Table 11: The experimental results of different multi-task optimization methods on NYUD-v2 with ResNet-18. The losses are weighted by homoscedastic uncertainty. Experiments are repeated over 3 random seeds and average values are presented. $\triangle_m \uparrow(\%)$ is used to indicate the percentage improvement in multi-task performance (MTP). The best results are expressed in **bold** numbers.

| Tasks | Depth | | SemSeg | | | Surface Normal | | | | | |
|---|---|---|---|---|---|---|---|---|---|---|---|
| | Distance (Lower Better) | | (%) (Higher Better) | | | Angle Distance (Lower Better) | | Within t degree (%) (Higher Better) | | | MTP |
| Method | rmse | abs_rel | mIoU | PAcc | mAcc | mean | median | 11.25 | 22.5 | 30 | $\triangle_m \uparrow(\%)$ |
| Independent | 0.659 | 0.183 | 34.46 | 65.51 | 46.50 | 23.36 | 16.67 | 34.89 | 62.19 | 73.33 | + 0.00 |
| GD | 0.608 | 0.158 | 39.02 | 69.29 | 51.48 | 22.06 | 15.47 | 37.98 | 65.01 | 75.68 | + 8.85 |
| MGDA | 0.623 | 0.162 | **39.43** | **69.30** | **51.79** | 22.65 | 15.77 | 37.39 | 64.03 | 74.66 | + 7.64 |
| PCGrad | 0.606 | 0.158 | 39.40 | 69.25 | 51.68 | 22.25 | 15.43 | 38.05 | 64.81 | 75.35 | + 9.04 |
| CAGrad | 0.600 | 0.156 | 38.62 | 68.74 | 51.03 | 22.27 | 15.43 | 38.11 | 64.85 | 75.32 | + 8.56 |
| Ours | **0.595** | **0.153** | 38.67 | 69.01 | 51.01 | **21.05** | **14.11** | **41.43** | **67.91** | **77.59** | **+ 10.61** |

### D.3 PASCAL-CONTEXT WITH HRNET-18

Table 12: The experimental results of different multi-task optimization methods on PASCAL-Context dataset with HRNet-18. The losses of all tasks are evenly weighted. Experiments are repeated over 3 random seeds and average values are presented. $\triangle_m \uparrow(\%)$ is used to indicate the percentage improvement in multi-task performance (MTP). The best results are expressed in **bold** numbers.

| Tasks | SemSeg | | PartSeg | Saliency | | Surface Normal | | | | | MTP |
|---|---|---|---|---|---|---|---|---|---|---|---|
| Method | (Higher Better) | | (Higher Better) | (Higher Better) | | Angle Distance (Lower Better) | | Within t degree (%) (Higher Better) | | | |
| | mIoU | PAcc | mIoU | mIoU | maxF | mean | median | 11.25 | 22.5 | 30 | $\triangle_m \uparrow(\%)$ |
| Independent | 60.30 | 89.88 | 60.56 | 67.05 | 78.98 | 14.76 | 11.92 | 47.61 | 81.02 | 90.65 | + 0.00 |
| GD | 61.65 | 90.14 | 58.35 | 65.80 | 78.07 | 16.71 | 13.82 | 39.70 | 75.18 | 87.17 | - 4.12 |
| MGDA | **63.52** | **90.68** | 60.38 | 64.99 | 77.57 | 17.00 | 14.13 | 38.58 | 74.47 | 86.77 | - 3.30 |
| PCGrad | 63.21 | 90.33 | 60.42 | 64.77 | 77.48 | 16.65 | 13.71 | 39.64 | 75.10 | 87.07 | - 2.90 |
| CAGrad | 63.44 | 90.53 | 60.11 | 64.83 | 77.52 | 16.92 | 13.98 | 39.03 | 75.01 | 86.92 | - 3.37 |
| Ours | 62.64 | 90.39 | **61.42** | **67.10** | **78.91** | **15.58** | **12.68** | **43.93** | **78.69** | **89.26** | **- 0.05** |

Table 13: The experimental results of different multi-task optimization methods on PASCAL-Context dataset with HRNet-18. The losses are weighted using Dynamic Weight Average (DWA). Experiments are repeated over 3 random seeds and average values are presented. $\triangle_m \uparrow(\%)$ is used to indicate the percentage improvement in multi-task performance (MTP). The best results are expressed in **bold** numbers.

| Tasks | SemSeg | | PartSeg | Saliency | | Surface Normal | | | | | MTP |
|---|---|---|---|---|---|---|---|---|---|---|---|
| Method | (Higher Better) | | (Higher Better) | (Higher Better) | | Angle Distance (Lower Better) | | Within t degree (%) (Higher Better) | | | |
| | mIoU | PAcc | mIoU | mIoU | maxF | mean | median | 11.25 | 22.5 | 30 | $\triangle_m \uparrow(\%)$ |
| Independent | 60.30 | 89.88 | 60.56 | 67.05 | 78.98 | 14.76 | 11.92 | 47.61 | 81.02 | 90.65 | + 0.00 |
| GD | **64.70** | **91.18** | 60.60 | 66.54 | 78.18 | 15.13 | 12.23 | 45.77 | 79.91 | 89.96 | + 1.02 |
| MGDA | 64.56 | 90.72 | 60.69 | 65.93 | 77.37 | 16.87 | 13.95 | 39.35 | 74.69 | 86.82 | - 2.17 |
| PCGrad | 64.35 | 90.98 | 60.99 | 66.12 | 77.65 | 15.92 | 13.11 | 41.98 | 76.21 | 88.03 | - 0.45 |
| CAGrad | 64.03 | 90.77 | 60.62 | 66.01 | 77.42 | 16.63 | 13.86 | 40.02 | 75.22 | 87.41 | - 1,98 |
| Ours | 63.89 | 90.73 | **61.89** | **67.39** | **79.08** | **14.94** | **12.10** | **46.27** | **80.57** | **90.41** | **+ 1.86** |

Table 14: The experimental results of different multi-task optimization methods on PASCAL-Context dataset with HRNet-18. The losses are weighted by homoscedastic uncertainty. Experiments are repeated over 3 random seeds and average values are presented. $\triangle_m \uparrow(\%)$ is used to indicate the percentage improvement in multi-task performance (MTP). The best results are expressed in **bold** numbers.

| Tasks | SemSeg | | PartSeg | Saliency | | Surface Normal | | | | | MTP |
|---|---|---|---|---|---|---|---|---|---|---|---|
| Method | (Higher Better) | | (Higher Better) | (Higher Better) | | Angle Distance (Lower Better) | | Within t degree (%) (Higher Better) | | | |
| | mIoU | PAcc | mIoU | mIoU | maxF | mean | median | 11.25 | 22.5 | 30 | $\triangle_m \uparrow(\%)$ |
| Independent | 60.30 | 89.88 | 60.56 | 67.05 | 78.98 | 14.76 | 11.92 | 47.61 | 81.02 | 90.65 | + 0.00 |
| GD | 64.40 | 91.05 | 62.28 | 68.13 | 79.64. | 14.95 | 12.14 | 46.19 | 80.36 | 90.34 | + 2.49 |
| MGDA | 64.04 | 90.88 | 61.18 | 67.65 | 79.23 | 15.02 | 12.20 | 45.93 | 80.02 | 90.11 | + 1.59 |
| PCGrad | **64.75** | **91.11** | **62.41** | 68.16 | 79.65 | 14.86 | 11.93 | 47.03 | 80.60 | 90.31 | + 2.85 |
| CAGrad | 64.01 | 90.77 | 61.32 | 67.55 | 79.01 | 15.08 | 12.31 | 45.87 | 79.98 | 90.05 | + 1.50 |
| Ours | 64.01 | 90.70 | 61.78 | **68.32** | **81.50** | **14.53** | **11.52** | **48.21** | **81.88** | **90.74** | **+ 2.90** |

# E  ADDITIONAL ABLATION STUDIES

**The order of updating tasks in Phase 1 has little impact on multi-task performance.** To learn task priority in shared parameters, Phase 1 updates each task-specific gradient one by one sequentially. To determine the influence of the order of tasks on optimization, we randomly chose 5 sequences of tasks and showed their performance in Table 15. From the results, we can see that the order of updating tasks in Phase 1 does not have a significant impact on multi-task performance.

Table 15: The experimental results for NYUD-v2 with HRNet-18 involved exploring different task sequence orders in Phase 1. We conducted ablation experiments with five randomly selected task sequences. Each task was represented by a single alphabet letter, as follows: S for semantic segmentation, D for depth estimation, E for edge detection, and N for surface normal estimation.

| Tasks | Depth | | SemSeg | | | Surface Normal | | | | | MTP |
|---|---|---|---|---|---|---|---|---|---|---|---|
| | Distance (Lower Better) | | (%) (Higher Better) | | | Angle Distance (Lower Better) | | Within t degree (%) (Higher Better) | | | |
| Method | rmse | abs_rel | mIoU | PAcc | mAcc | mean | median | 11.25 | 22.5 | 30 | $\triangle_m \uparrow (\%)$ |
| Independent | 0.667 | 0.186 | 33.18 | 65.04 | 45.07 | 20.75 | 14.04 | 41.32 | 68.26 | 78.04 | + 0.00 |
| N-D-S-E | 0.574 | 0.157 | 41.12 | 70.44 | 53.77 | 19.60 | 12.52 | 46.01 | 71.33 | 80.02 | + 14.47 |
| D-S-N-E | 0.568 | 0.153 | 40.92 | 70.23 | 53.56 | 19.55 | 12.47 | 46.09 | 71.50 | 80.12 | + 14.65 |
| E-D-S-N | 0.568 | 0.150 | 40.97 | 70.22 | 53.59 | 19.58 | 12.50 | 46.08 | 71.44 | 80.07 | + 14.65 |
| D-N-E-S | 0.571 | 0.153 | 41.03 | 70.31 | 53.68 | 19.49 | 12.44 | 46.17 | 71.58 | 80.17 | + 14.71 |
| S-D-E-N | 0.565 | 0.148 | 41.10 | 70.37 | 53.74 | 19.54 | 12.45 | 46.11 | 71.54 | 80.12 | + 15.00 |

**the speed of learning the task priority differs based on the convolutional layer's position.** Phase 1 establishes the task priority during the initial stages of the network's optimization. Meanwhile, Phase 2 maintains this learned task priority, ensuring robust learning even when the loss for each task fluctuates. However, The timing at which task priority stabilizes varies based on the position of the convolutional layer within the network, as illustrated in Fig. 5. This may suggest that optimizing by wholly separating each phase could be inefficient.

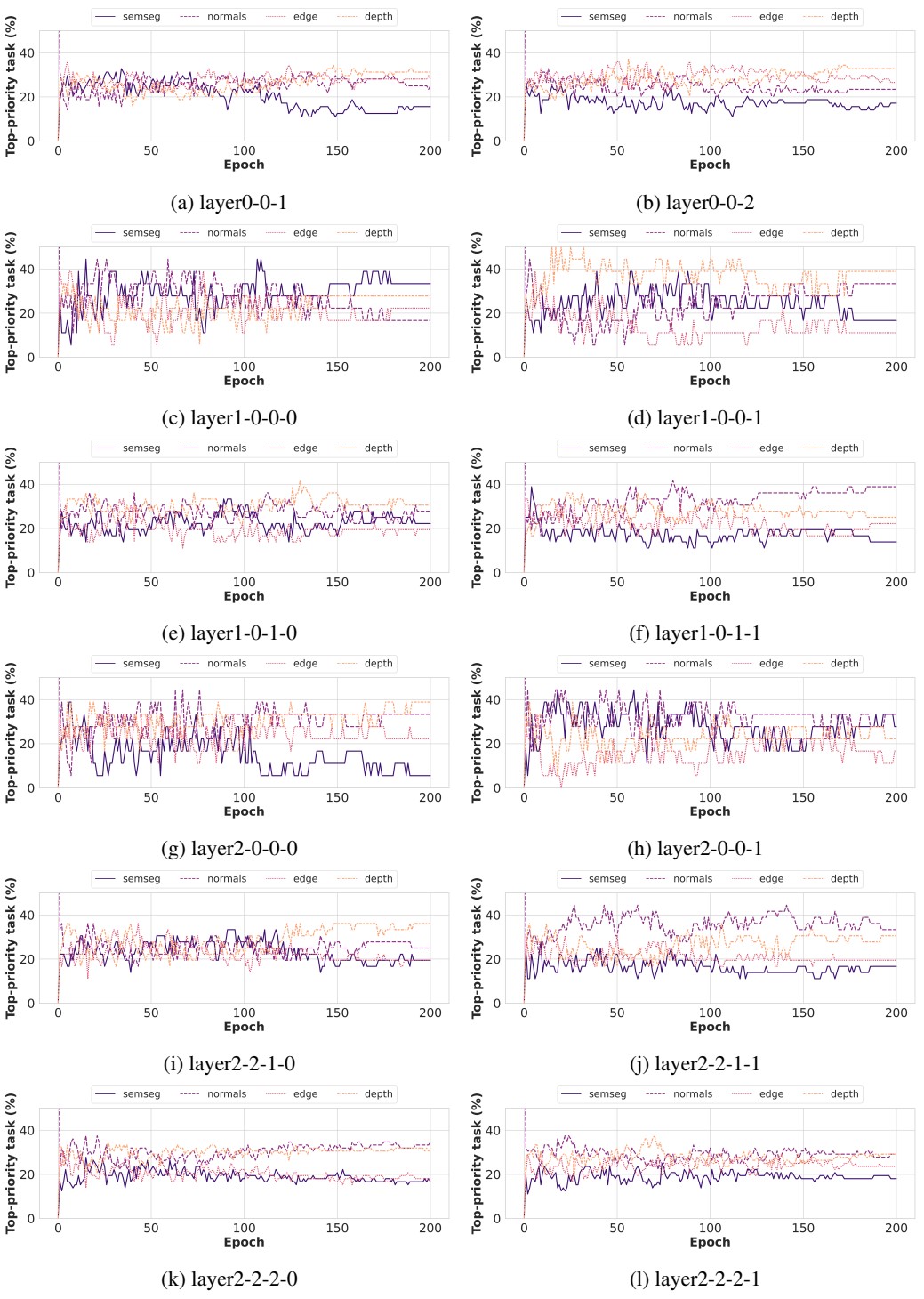

Figure 5: Visualization of the percentage of top-priority tasks over training epoch depending on the position in the network. We randomly selected several convolutional layers from the Network. The timing at which task priority stabilizes varies depending on the position of the convolutional layer.

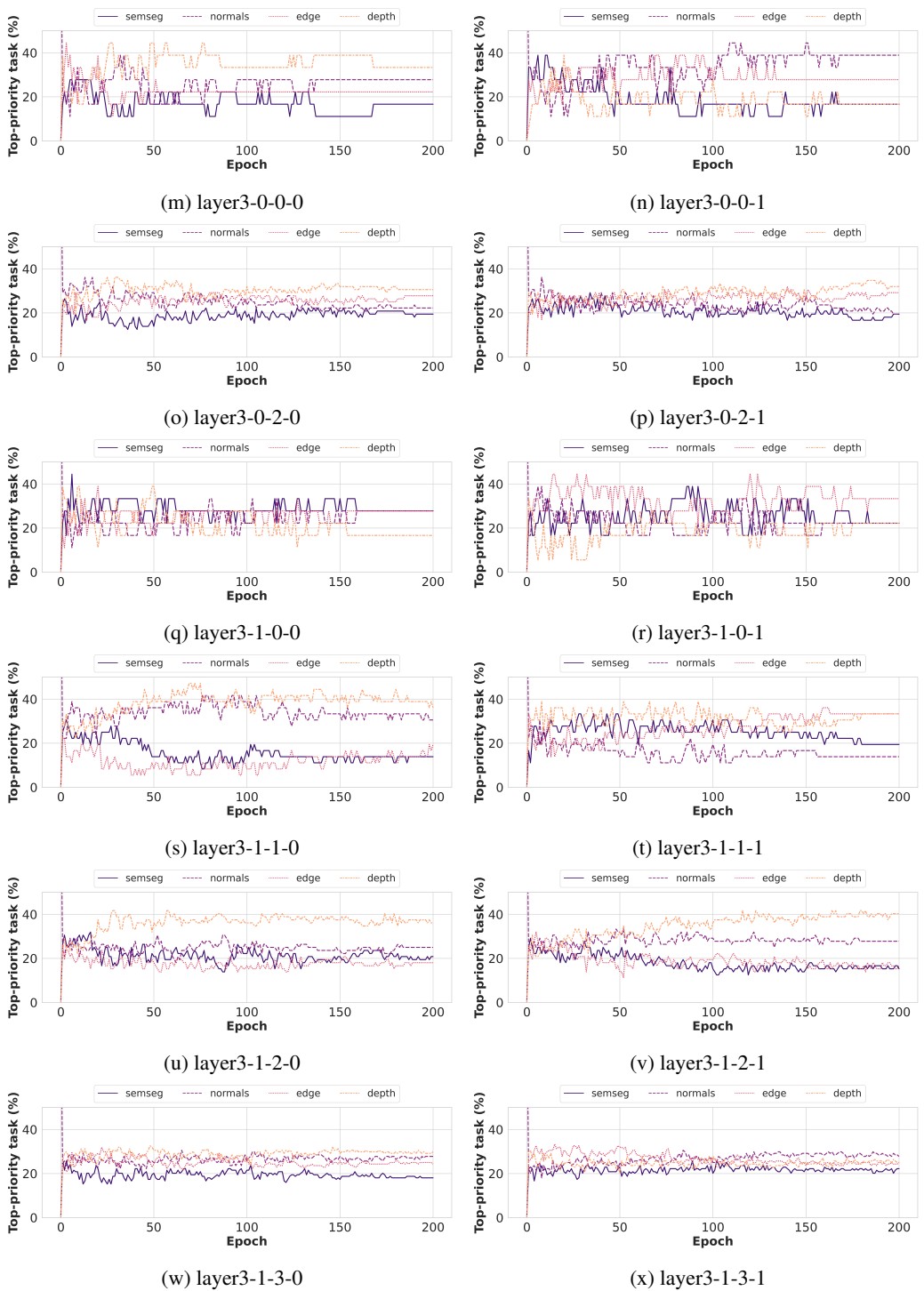

Figure 5: Visualization of the percentage of top-priority tasks over training epoch depending on the position in the network. We randomly selected several convolutional layers from the Network. The timing at which task priority stabilizes varies depending on the position of the convolutional layer.

