# OpenReview forum: "Connection Strength-Based Optimization for Multi-Task Learning"
_ICLR.cc/2024/Conference — ICLR 2024 Conference Withdrawn Submission_

### Official Review · Reviewer_twaP · 2023-10-22

**Soundness:** 2 fair
**Presentation:** 3 good
**Contribution:** 2 fair
**Rating:** 3
**Confidence:** 4

**Summary:**

This work tackles the problem of negative transfer when training multi-task learning models. The authors identify that it is important to find task priority and use connection strength as a measure for it. They propose a two-phase optimization algorithm to ensure gradient update favors tasks with high priority. Empirical evaluations suggest good performance of the proposed algorithm.

**Strengths:**

1. The empirical performance is strong. It is rare to see scenarios where one algorithm simultaneously outperforms others on all tasks evaluated (i.e., Pareto improvement).

2. The problem identified by the authors is important, as other gradient manipulation algorithms do not consider the priority of tasks.

**Weaknesses:**

1. The description of implicit connections is very vague. The authors only say that each connection is different for each task, but what is the connection? No proper definition is given.

2. The phase I operation is confusing. How is the step-by-step gradient update related to learning task priority?

3. Phase I requires extra computational overhead which potentially cannot scale to larger scale problems. See question 3.

4. The claim of achieving a new Pareto front is unreasonable. By definition, as long as the network architecture is fixed, the Pareto front for the optimization problem is also fixed. As far as I can recall, both uniform scalarization, MGDA and CAGrad provably converge to Pareto optimal solutions. In that case, how could the proposed algorithm achieve Pareto improvement over them?

**Questions:**

1. What is the usage of batch normalization? Is it used to eliminate the effect of different loss scales?

2. In the discussion of the algorithm, the connection strength is only discussed in CNN. Is it architecture agnostic or specific to CNN?

3. From my understanding, in each iteration, if there are k tasks, phase I operation requires forward passing the data k times because the parameters need to be updated for each task. However, other MTL methods only require 1 pass. Could you provide some discussion about this computation overhead?

4. The idea of prioritizing the task whose gradient leads to the largest decrease for the joint loss is explored before [1]. What is the connection between the proposed method and theirs? [1] provides a simpler and solves task priority explicitly, instead of using heuristics like connection strength.

[1] Focus on the Common Good: Group Distributional Robustness Follows. Piratla et al. ICLR 2022.

---

### Official Review · Reviewer_PR9H · 2023-10-31

**Soundness:** 3 good
**Presentation:** 3 good
**Contribution:** 3 good
**Rating:** 6
**Confidence:** 3

**Summary:**

In this paper, the authors introduce a novel algorithm that can find better Pareto optimal solutions to multi-task learning (MTL) problems, in terms of the final loss obtained for each individual task loss function. To achieve this, the authors introduce a concept of task priority, and propose to learn the task priorities in a given MTL network and then update MTL network while preserving this task priority. Authors provide some theory to justify the importance of task priority and how preserving task priority lead to better Pareto optimal solutions, and propose a practical implementation for identifying task priorities in a MTL network using the notion of connection strengths in the network. Authors implement the proposed method in several MTL benchmarks and provide empirical validation for the usefulness of the proposed method.

**Strengths:**

* The idea identifying and preserving task priorities in MTL models and how it relates to better Pareto optimal points is interesting and seems to have potential to improve MTL performance.

* Authors provide some theoretical justification and convincing empirical justification to validate the proposed method.

* The flow of the paper is easy to follow.

**Weaknesses:**

* The proposed method involves multiple gradient updates and connection strength calculations (either one at a given iteration), which seems very computationally demanding compared to existing gradient based MTL methods like MGDA (which are already computationally intensive).

* Although the authors mention the proposed method of priority allocation “prevent a specific task from exerting dominant influence over the entire network”, it is not easy to see why this is the case. The definition of the task priority does not seem to exclude the case where a specific task exerting dominant influence over the entire network.

* The y axis of Figure 4, “percentage of top-priority tasks” is ambiguous. What is the relationship between this quantity and the connection strength defined in Eq. 7?

Minor comments:

* In Algorithm1, using $p$ to denote the phase mixing probability and the dummy variable in the inner loop in Phase 2 might be confusing

**Questions:**

* How does the proposed method compare with prior MTL methods in terms of computation intensity/time consumption?

* Does the order of task updates in phase 1 affect task priority assigning for parameters?

* Is there a specific order in which gradient projection is done in phase 2?

---

### Official Review · Reviewer_8sTu · 2023-11-01

**Soundness:** 3 good
**Presentation:** 3 good
**Contribution:** 2 fair
**Rating:** 5
**Confidence:** 4

**Summary:**

In this submission, the authors present a novel optimization algorithm for muti-task learning named connection strength-based optimization. This framework can help us find which portions of the parameters are important and boost the multi-task performance. The authors also conduct numerical results to show the good performance of this proposed framework.

**Strengths:**

This submission is well-organized with clear language and structures. The authors gave detailed description for the proposed algorithms. They also conduct a lot of numerical experiments on large datasets and these empirical results are pretty good compared with some state-of-art optimization methods. The idea is pretty interesting and enlightens some promising future direction for the optimization community.

**Weaknesses:**

However, this submission has one significant disadvantage. That is the lack of theoretical analysis for the proposed algorithms. Although the authors gave a lot of details regarding the algorithm itself and a lot of intuitions and implementation for the proposed method. There is no theoretical analysis or convergence results for this method. Although the empirical results from the numerical experiments are very promising compared to other methods. This lack of theoretical analysis make the contribution of this submission not significant enough.

**Questions:**

Please check the weakness section.

---

### Official Review · Reviewer_3sc7 · 2023-11-05

**Soundness:** 3 good
**Presentation:** 4 excellent
**Contribution:** 3 good
**Rating:** 6
**Confidence:** 4

**Summary:**

This paper proposes a novel multi-task optimization method which is motivated by the idea that different sets of parameters are "prioritized" for different tasks. The authors conjecture that, for each task, there exists a subset of parameters for whose "priority" on that task (measured by the effect of the task gradient of the subset on the overall loss) is greater than other tasks, and that conflict should be mitigated in the parameter subset according to the priority task. This is different from prior multi-task optimization methods, which treat the each parameter uniformly when mitigating task conflict.

Because determining this subset exactly is intractable for reasonably sized models, the authors propose instead to use "connection strength", measured at the output of each channel of a CNN, to determine the task priority for the subset of parameters associated with that channel. The explicit connection strength metric is determined by the coefficients of that output channel learned by a task-specific batch-norm combined with the sum of the weights of the parameter subset that apply to that output channel.

Leveraging this notion of connection, the authors propose a two-phase optimization process which learns "implicit connections" then maintains them explicitly. The implicit phase essentially treats the multi-task learning process as a round-robin style learning, rather than aggregating the gradients of each task at each step. The second phase behaves on aggregated gradients over each task, and mitigates conflict by first identifying task priority via explicit connection strength, and then projecting all task gradients in each subset to it's priority task. These two phases are used in an annealing fashion, such that early on the "implicit phase" is used more often and towards the end of training, the "explicit phase" is used more often to maintain connection strength.

The authors demonstrate the effectiveness of their method on NYUD-v2, Cityscapes, and PASCAL, 3 common multi-task methods, and find that their method outperforms the baselines they consider (MGDA, PCGrad, CAGrad) consistently, and under different loss-scaling methods, and show their method achieves better pareto optimality than prior methods.

The also perform an ablation study, showing that the "explicit" phase is important for final model generalization even though using solely the "implicit phase" achieves lower training loss, highlighting the importance of maintaining connection strength.

**Strengths:**

- At a high level, the method makes a lot of sense; it is intuitive that subsets of the parameter space should consider mitigating conflict across tasks differently, depending on their influence to each task and that these subsets might not be handled naturally by aggregating gradients.
- The paper is relatively well-written - the motivation and overall method is clear, and the results are presented well.
- The method is simple, serving almost as an extension of PCGrad to subsets, and therefore easy to understand.
- The results on the considered datasets are very strong - the method is shown to improve performance on many architectures and weight-balancing schemes, which are generally not shown for new multi-task optimization methods.
- The paper considers an ablation on the different phases of optimization, and demonstrates the necessity of the explicit connection phase for generalization.

**Weaknesses:**

- The notion of explicit connection strength is only defined for CNN channels. While it seems intuitive that some similar extension could be reached for e.g. transformers heads, the extension to fully connected layers is less clear. Moreover, the splitting by channel parameters seems somewhat artificial and the limitations are not discussed.
- Some parts of the paper are a bit unclear, or leave important ideas underspecified. For instance, I found the idea of "implicit connection" hard to grasp. My current understanding is that it is the same as explicit connection, but it emerges implicitly during training rather than being explicitly optimized. However, the initial portion of the paper left me understanding them as two distinct definitions of connection. Also, see question.
- The paper doesn't consider the obvious baseline which mixes phase 1 with PCGrad. This would demonstrate the necessity of applying PCGrad to the different subsets of parameters separately, while controlling for the fact that the model benefits from a different training strategy early on (round-robin style training, rather than fully aggregated gradients like the other baselines).

**Questions:**

- Is connection strength order determined mainly by the task-specific batchnorm coefficients? What is the purpose of multiplying the weights of the channel parameters into the equation, if they are constant for all tasks?
-